# Magnon detection using a ferroic collinear multilayer spin valve

Joel Cramer[1,2], Felix Fuhrmann[1], Ulrike Ritzmann[1,3], Vanessa Gall[3], Tomohiko Niizeki[4], Rafael Ramos [4], Zhiyong Qiu [4,8], Dazhi Hou[4], Takashi Kikkawa [4,5], Jairo Sinova[1], Ulrich Nowak[3], Eiji Saitoh[4,5,6,7] & Mathias Kläui [1,2]

Information transport and processing by pure magnonic spin currents in insulators is a promising alternative to conventional charge-current-driven spintronic devices. The absence of Joule heating and reduced spin wave damping in insulating ferromagnets have been suggested for implementing efficient logic devices. After the successful demonstration of a majority gate based on the superposition of spin waves, further components are required to perform complex logic operations. Here, we report on magnetization orientation-dependent spin current detection signals in collinear magnetic multilayers inspired by the functionality of a conventional spin valve. In $Y_3Fe_5O_{12}|CoO|Co$, we find that the detection amplitude of spin currents emitted by ferromagnetic resonance spin pumping depends on the relative alignment of the $Y_3Fe_5O_{12}$ and Co magnetization. This yields a spin valve-like behavior with an amplitude change of 120% in our systems. We demonstrate the reliability of the effect and identify its origin by both temperature-dependent and power-dependent measurements.

[1] Institute of Physics, Johannes Gutenberg-University Mainz, 55099 Mainz, Germany. [2] Graduate School of Excellence Materials Science in Mainz, 55128 Mainz, Germany. [3] Department of Physics, University of Konstanz, 78457 Konstanz, Germany. [4] Advanced Institute for Materials Research, Tohoku University, Sendai 980-8577, Japan. [5] Institute for Materials Research, Tohoku University, Sendai 980-8577, Japan. [6] Center for Spintronics Research Network, Tohoku University, Sendai 980-8577, Japan. [7] Advanced Science Research Center, Japan Atomic Energy Agency, Tokai 319-1195, Japan. [8] Present address: School of Materials Science and Engineering, Dalian University of Technology, Dalian 116024, China. Correspondence and requests for materials should be addressed to M.Käu. (email: klaeui@uni-mainz.de)

Almost three decades ago, the discovery of the giant magnetoresistance effect (GMR)[1,2] marked a key moment for the research field of spintronics based on spin-polarized charge currents. The inclusion of the spin degree of freedom in information technology promises, among others, the implementation of logic devices with increased speed or capacity as compared to conventional CMOS electronics[3]. This approach has been driven to the next level by the field of magnon spintronics[4]. Magnons are quasiparticles (quanta) of the collective excitation of electron spins (spin waves) in magnetically ordered systems. Because of being spin-1 particles, a great number of magnons with equal propagation direction yields a pure spin current (SC) carrying information in the form of angular momentum. One of the advantages of spin wave-mediated information transport, especially in the case of insulating magnetic oxides, is the absence of electron motion and thus the absence of power dissipation due to Joule heating during the transport.

Consequently, in view of not only new physical phenomena but also considerable application potential, the investigation of generation and detection of pure magnonic SCs in insulators has attracted significant attention in recent years. One of the most prominent systems for such SCs is the insulating ferrimagnet (FM) $Y_3Fe_5O_{12}$ (YIG). Inherently, YIG exhibits the lowest measured Gilbert damping constant $\alpha \propto 10^{-5}$, enabling long-distance spin propagation and thus efficient transport of spin information[5,6]. In addition to other techniques, the detection of magnonic SCs can be achieved most conveniently by means of the inverse spin Hall effect (ISHE)[7]. Likewise, for the generation a number of approaches have been developed: ferromagnetic resonance (FMR) spin pumping[8–11], thermal generation of SCs induced by the spin Seebeck effect (SSE)[12–14], and the electrical injection of magnons by the spin Hall effect[15,16].

In recent years, significant progress has been reported with respect to the concepts and implementations of magnon-based logic operations. It was shown, for instance, that by localized Oersted fields spin wave propagation paths can be manipulated, yielding a de-multiplexer like behavior[17]. The amplitude of propagating spin waves, then again, can be modified by employing magnonic crystals that allow one to introduce attenuation effects comparable to the functionality of a transistor[18]. Most recently, it was shown that the superposition of coherent or incoherent magnons in a spin wave bus or a non-local geometry, respectively, enables the implementation of a fully functional logic majority gate[19–22]. These concrete demonstrations of magnon logic highlight the potential of this new information-processing scheme, implying the necessity for further logic building blocks to accomplish more complex operations. While the above-mentioned approaches concentrate on the manipulation of SCs in the propagation phase, an extremely advantageous functionality, especially with regard to scalability, would be the integration of a switch-like device that enables a magnetization alignment-dependent local SC detection signal amplitude.

In this work, we demonstrate the magnetization orientation-dependent detection of magnonic SCs in YIG|CoO|Co spin valve structures. By means of FMR spin pumping[8–11], a pure magnonic SC is emitted from the YIG into the CoO layer, which is eventually detected via the ISHE[7] in Co as a voltage signal. A distinct difference in the signal amplitude of up to 120% is observed for different relative alignments of the YIG and Co magnetization, which are either parallel or antiparallel. The comparison to atomistic spin dynamics simulations shows that the results obtained cannot be explained by a purely magnonic spin valve effect, but that electronic effects, as, for instance, a spin-dependent ISHE, have to be considered. In addition to the spin-pumping signal, a further voltage signal is observed. Its sign depends on the magnetization direction of Co and, from temperature-dependent and power-dependent measurements, it is concluded to be dominated by anomalous Hall effect (AHE) induced spin rectification (SR)[23–26].

## Results

**Experiment**. Inspired by the magnetoresistive spin valve concept, we investigate the magnetization alignment-dependent detection of magnonic SCs in ferroic collinear YIG|CoO|Co structures, to which in the following we refer as magnon spin valve effect. The sample layout used in this work is illustrated in Fig. 1a. In this system, YIG serves as a SC source driven by FMR. The coherent rotation of the YIG magnetization emits a pure SC at the YIG|CoO interface into the CoO layer. The Co layer is used as an active switchable layer, as well as for the subsequent SC detection. As previously shown, ferromagnets as well exhibit an ISHE allowing for the efficient conversion of a SC into a charge current[27,28]. The insulating antiferromagnet (AFM) CoO was chosen as interlayer material as it decouples the ferromagnetic layers, while simultaneously allowing for transport of the pumped SC[29–33] by diffusive AFM magnons[34,35] or evanescent modes[36]. Furthermore, in its antiferromagnetic phase CoO introduces exchange biasing of the Co film. Besides a unidirectional anisotropy that shifts the Co hysteresis loop, an additional uniaxial anisotropy yields an enhanced coercive field at which the Co magnetization switches[37,38]. The enhanced coercivity allows one to perform spin pumping measurements with parallel and antiparallel alignment of YIG and Co at the FMR resonance field. For a good signal-to-noise ratio $H_{FMR}^{YIG} \gtrsim 800$ Oe is necessary, and such a high switching field is unattainable for thin Co films without exchange biasing. Both the CoO and the Co layer thickness were varied and here we concentrate on the results of three typical stacks. For reasons of simplification, in the following the investigated trilayers YIG (5 μm)|CoO (2 nm)|Co (4 nm), YIG (5 μm)|CoO (3 nm)|Co (4 nm), and YIG (5 μm)|CoO (5 nm)|Co (6 nm) will be referred to as sample A, B, and C, respectively.

**Spin-thermoelectric measurements**. To check if the magnetic switching fields of Co are sufficiently high to carry out FMR spin pumping with a good signal-to-noise ratio ($H_{FMR}^{YIG} \gtrsim 800$ Oe), these are determined by temperature-dependent magneto-galvanic measurements. The anomalous Nernst effect (ANE) occurring in the Co layer, which is generated by an out-of-plane temperature gradient, yields a signal proportional to the in-plane magnetization component and thus we can detect the Co switching. Common magnetometry (e.g., SQUID) yields the YIG switching due to the much smaller volume of the Co film as compared to the YIG layer. A typical ANE hysteresis curve for sample A recorded at $T = 120$ K is shown in Fig. 1b. The graph reveals coercive fields of $H_c^+$ [$H_c^-$] $\simeq 1250$ Oe [$-970$ Oe] and large squareness with $V_{ANE}^{H=0}/V_{ANE}^{sat} \simeq 0.95$, implying that Co switches without extensive formation of magnetic domains. Note that in contrast to previous reports[32], the thermal SC of the YIG film generated by the SSE[12,39] was not observed within the experimental resolution. A potential explanation for this may be given by different CoO thicknesses and structures grown on different YIG films (polycrystalline vs. single crystalline)[40], and the smaller ISHE in Co compared to the previously used Pt. In addition, the absence of a detected spin Seebeck signal rules out pinholes in the CoO layer.

**Spin signal in YIG|CoO|Co excited by spin pumping**. Having established the magnetization reversal properties of both YIG and Co, we study the spin signal as a function of alignment of the YIG and Co layer. Figure 1c–f shows field-dependent voltage signals generated in sample A at $T = 120$ K induced by $f = 4.5$ GHz

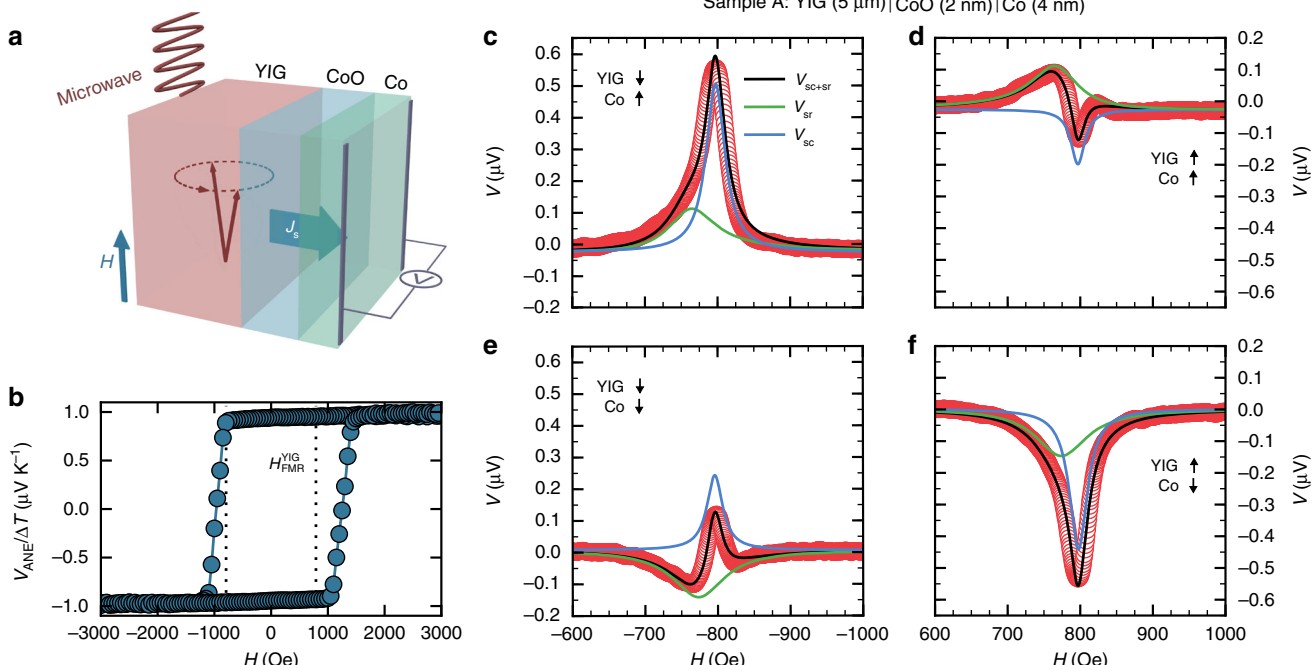

**Fig. 1** Measurement concept and spin pumping voltage responses. **a** Illustration of the investigated $Y_3Fe_5O_{12}$ (YIG)|CoO|Co tri-layer system. A spin current $J_s$ induced in the YIG by ferromagnetic resonance (FMR) spin pumping propagates through the CoO intermediate layer and is detected electrically in the Co layer via the inverse spin Hall effect (ISHE). The relative magnetization orientation of YIG and Co can be adjusted to be either parallel or antiparallel. **b** Typical cobalt anomalous Nernst effect (ANE) hysteresis loop observed for sample A at $T = 120\,K$, being field-cooled at $H_{ext} = 90\,kOe$. Coercive fields of the Co layer of $H_c^+ [H_c^-] \simeq 1250\,Oe\ [-970\,Oe]$ and an elevated squareness $V_{ANE}^{H=0}/V_{ANE}^{sat} \simeq 0.95$ are observed. **c–f** Field-dependent voltage signals detected in sample A induced by microwave irradiation ($f = 4.5\,GHz\ P_{abs} \approx 48\,mW$, and $T = 120\,K$). The total signal is given by a superposition of two distinct signals $V_{sc}$ and $V_{sr}$; the sign of $V_{sc}$ ($V_{sr}$) depends on the magnetization orientation of YIG (Co). While $V_{sc}$ is the spin transport signal of the spin current $J_s$ induced in the YIG film, $V_{sr}$ inherently originates from the Co layer, potentially due to heating or spin rectification effects. The amplitude of $V_{sc}$ depends on the relative alignment of YIG and Co. Microwave absorption spectra (see Supplementary Note 1) show that FMR of YIG is excited at this frequency at $H_{FMR}^{YIG} = \pm788\,Oe$

microwave irradiation (measurements at different resonance frequencies did not yield any significant change of the qualitative signal). In all cases the microwave power absorbed by the sample accounts for $P_{abs} \approx 48\,mW$. By the application of specific field sweep sequences, parallel (d, e), as well as antiparallel (c, f) alignment of YIG and Co is realized. In the parallel state (Fig. 1d, e) a multi-peak voltage signal appears. Considering possible combinations of symmetric, as well as asymmetric terms, the signal is fitted well by two overlapping Lorentzian line shapes of opposite sign, slightly shifted peak field values, and different line widths. The antiparallel state (Fig. 1c, f), on the other hand, exhibits a voltage peak of one polarity but with a significant asymmetry, which again can be fitted by two overlapping Lorentzian functions. The comparison of all four datasets allows us to both separate and attribute the peaks to different effects. Depending on the individual orientation of YIG and Co, the peaks change sign separately and thus can be identified as (i) the signal of the SC generated by spin pumping from the YIG, transmitted across the CoO, and detected by the ISHE in the Co (blue curves in Fig. 1c–f) and (ii) a second signal that depends only on the Co layer direction and thus originates from the Co (green curves in Fig. 1c–f). We start by exploring the latter, for which possible explanations are a thermally induced ANE signal due to dynamic microwave heating[41] or an AHE-induced SR signal[23–26]. The microwave driving current $I_{rf}$ partially flows through the Co layer by means of capacitive coupling and, together with the out-of-plane component of the Co magnetization precession excited by dipolar-coupled YIG magnon modes[26], results in a rectified dc voltage. While both effects can contribute,

temperature-dependent measurements support the SR mechanism as discussed below. Note that in metallic ferromagnets directly driven by FMR, the SR effect yields both a symmetric and an asymmetric signal shape around the resonance field, of which the latter is due to a phase shift between the rf current flow and the magnetization dynamics induced[25]. In this work, we find a dominating symmetric contribution to the signal, while only a negligible asymmetric contribution to the Co signal is seen, which can thus be neglected in the analysis. A potential explanation is the cobalt being driven off-resonance and coupling toward multiple YIG magnon modes, which may result in a weak field dependence of the phase between the coupled rf current and the Co magnetization rotation. Regardless of its origin, as this signal depends only on the Co layer direction, it can be easily distinguished from the SC transport signal, which is the main thrust of this work. In the following we denote the signal that depends on the Co orientation as $V_{sr}$ (green), while the signal that depends on the relative orientation of the two layers is denoted as $V_{sc}$ (blue).

While the Co layer-dependent effect is based on known mechanisms, the actually intriguing discovery in this work is the alignment-dependent amplitude of $V_{sc}$. Whereas $V_{sr}$ does not depend on the relative alignment of the layers, the amplitude of $V_{sc}$ is nearly twice as large in the antiparallel alignment state as compared to the parallel state ($V_{sc}^{\uparrow\downarrow} > V_{sc}^{\uparrow\uparrow}$). Qualitatively the same behavior is observed for sample B and C (see Supplementary Note 1). Furthermore, we find that changing the polarity of the external field while field-cooling the sample ($H_{ext} = \pm90\,kOe$) yields no significant difference in both the microwave absorption

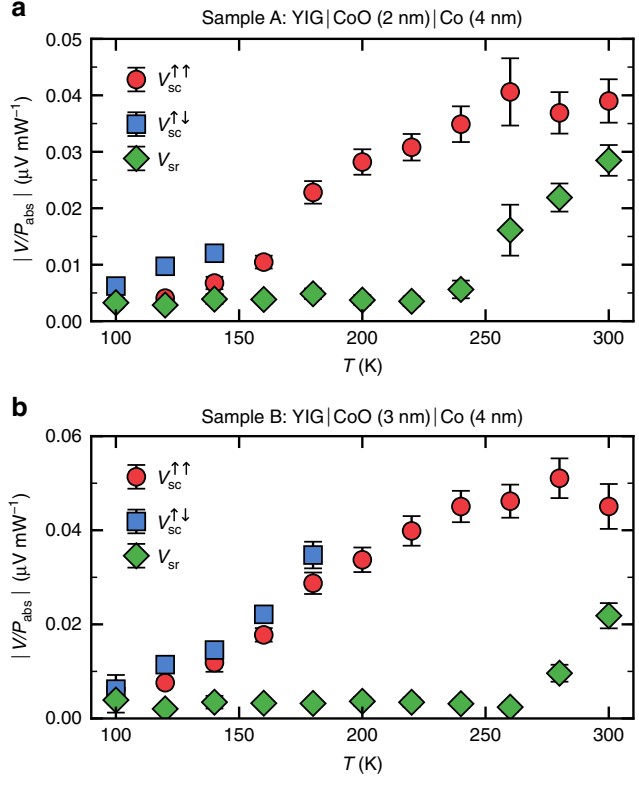

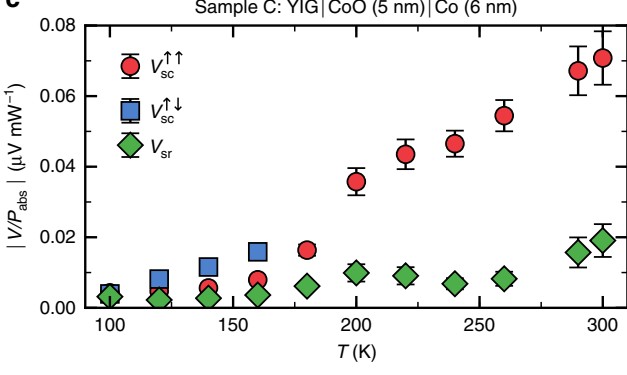

**Fig. 2** Temperature dependence of voltage signal amplitudes. **a–c** Amplitudes of $V_{sc}$ for parallel and antiparallel alignment and $V_{sr}$ normalized by the absorbed microwave power as a function of temperature for **a** sample A, **b** sample B, and **c** sample C. Antiparallel alignment of $Y_3Fe_5O_{12}$ (YIG) and Co is observed until a critical, sample-dependent temperature. The temperature range from 100 to 300 K was chosen as below 100 K a decreased signal-to-noise ratio impedes straightforward data analysis, whereas higher temperatures were avoided to prevent possible undesirable degradation of the multilayer stack. The error bars are calculated by error propagation including errors of the spin signal amplitude and absorbed power amplitude, as well as a systematic error estimated by the measurement device accuracy

spectrum and the detected voltage signal. This implies a negligible impact of any exchange bias effect at the CoO|Co interface, as well as of any potential exchange spring formed at the YIG|CoO interface on the observed effects.

**Origin of the spin signals in the YIG|CoO|Co magnon spin valve**. To investigate the origin of both signals, temperature-dependent spin pumping measurements were performed, see

Fig. 2a–c. The SC signal $V_{sc}$ can be measured up to a critical temperature for both the parallel and the antiparallel state, while for temperatures above this temperature only the parallel alignment can be established. This critical temperature depends on the investigated stack and is due to the fact that above a certain temperature $H_{FMR}^{YIG}$ exceeds the coercive field of the Co layer, preventing FMR for antiparallel alignment of the layers (for details, see Supplementary Note 2). For the accessible temperature range with antiparallel alignment, we always find $V_{sc}^{\uparrow\downarrow} > V_{sc}^{\uparrow\uparrow}$.

In general, the amplitude of $V_{sc}$ increases with increasing temperature. Furthermore, in the case of samples A and B a signal maximum is seen at temperatures between 250 and 300 K. This behavior can be explained by an enhanced spin conductivity of the CoO layer near its antiferromagnetic-paramagnetic phase transition[33]. It was previously shown that in thin CoO films $T_{Néel}$ is below its bulk value $T_{Neel}^{bulk} = 290$ K, due to finite size effects and decreases with decreasing film thickness[42,43]. Here, temperature-dependent ANE measurements for sample A (see Supplementary Note 2) reveal an exchange bias blocking temperature of $T_B \approx 250$ K, suggesting $250$ K $\leq T_{Néel} \leq 290$ K[43] for this specific stack. In the other multilayers with thicker CoO, $T_{Néel}$ is higher, which explains the monotonic increase of $V_{sc}$ for $d_{CoO} = 5$ nm up to 300 K.

For the amplitude of $V_{sr}$, the observed temperature dependence is qualitatively different. Initially, with increasing temperature $V_{sr}$ remains constant until it starts to increase significantly above a specific temperature, which is characteristic for each stack. Based on comparison with the Co switching we see that for sample A this temperature coincides with the blocking temperature $T_B$, above which the anisotropy introduced by exchange biasing vanishes. With regard to the possible mechanism of generating $V_{sr}$, the disappearance of this additional anisotropy entails a weakened cobalt magnetization precession damping. The magnetization consequently precesses at a larger cone angle[44], yielding a larger AHE voltage. The temperature above which the amplitude of $V_{sr}$ starts to increase is higher for thicker CoO, as expected for a thickness-dependent phase transition temperature. Finally, the observed temperature dependence supports the conclusion that $V_{sr}$ is not dominated by a magneto-galvanic effect as one would not expect a sudden onset of the signal above 250 K for the ANE.

Beyond the temperature-dependent data, further information about the signal origin can be obtained from the applied microwave power dependence. In Fig. 3a–c, we show the voltage amplitudes of $V_{sc}$ and $V_{sr}$ as a function of the absorbed microwave power. The SC transport amplitude exhibits a nonlinear power dependence, regardless of the alignment state. The onset of a saturation effect is indicated at elevated microwave powers, a behavior that was observed before for spin pumping in YIG/Pt structures[45], showing that this signal scales with the generated SC strength in the YIG. The power dependence of $V_{sr}$, on the other hand, does not allow for an equally simple interpretation. According to Azevedo et al.[24] one expects a linear power dependence for $V_{sr}$ due to its proportionality to the dynamic field driving the Co precession. While the power dependence observed is in line with this explanation, other possible mechanisms that yield a linear power dependence cannot be excluded.

Finally, we note that the fact that the peak fields for the two effects are slightly different is expected from their different origins. The peak of the SR signal will be at the field of the maximum of the excited magnon modes in the 5 μm thick YIG crystal that couples to the Co magnetization via dipolar exchange[26]. The peak of the SC transmission signal, however, will be at the field that corresponds to the maximum generation of SC pumped into the CoO at the YIG–CoO interface. Since the

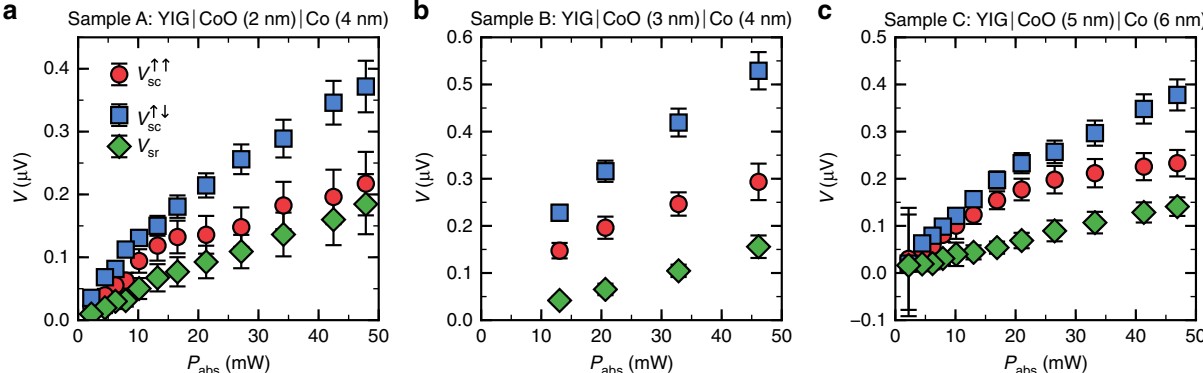

**Fig. 3** Power dependence of voltage signal amplitudes. **a–c** Spin pumping and spin rectification amplitude as a function of the absorbed microwave power for **a** sample A, **b** sample B, and **c** sample C. Whereas the spin rectification amplitude ($V_{sr}$) exhibits a rather linear power dependence, the spin pumping amplitude ($V_{sc}$) follows a non-linear power dependence for both magnetization alignment states of $Y_3Fe_5O_{12}$ (YIG) and Co, indicating an incipient saturation. The frequency of the applied microwave is $f = 4.5$ GHz, the system temperature is $T = 120$ K. The error bars are calculated by error propagation including errors of the spin signal amplitude, as well as a systematic error estimated by the measurement device accuracy

latter depends on the resonance at the interface while the former covers more of the bulk volume, the observed slight difference is not surprising.

**Magnon spin valve effect**. The alignment-dependent signal amplitude of SC detection now naturally lends itself to the implementation of a magnon spin valve effect. Comparing the amplitude of $V_{sc}$ for parallel and antiparallel alignment in Fig. 1, we find an effect amplitude of $(V_{sc}^{\uparrow\downarrow} - V_{sc}^{\uparrow\uparrow})/V_{sc}^{\uparrow\uparrow} = 120\%$ for sample A at $T = 120$ K. To check the reliability of the magnon spin valve effect, the magnetization direction of the Co top layer is switched several times in a row by driving the external field above (below) the Co coercive field $H_c^+ = 1250$ Oe ($H_c^- = -970$ Oe). Subsequent to every switching event, the voltage response toward the applied microwave is recorded at the FMR resonance field of YIG at FMR $H_{FMR}^{YIG} = 788$ Oe, see Fig. 4. Since in typical application schemes the external field is fixed instead of being swept and, thus, one voltage level is probed instead of acquiring the whole absorption spectrum, we choose a single fixed field value at which we acquire the signal amplitude. We find a large difference in the signal difference of the total voltage for parallel and antiparallel alignment. For sample A, at $H \approx H_{FMR}^{YIG}$ an absolute voltage difference of 408 nV and a spin valve effect amplitude of $\left(V_{sc+sr}^{\uparrow\downarrow} - V_{sc+sr}^{\uparrow\uparrow}\right)/V_{sc+sr}^{\uparrow\uparrow} = 290\%$ is found. The presented data clearly demonstrate that switching the Co layer yields reliably different and reproducible voltage levels with little scatter. No distinguishable reduction of the voltage difference was observed for days of measurements of switching, which signifies reliable long term operation without having to reset the magnetization. Furthermore, the size of the signal change is only weakly temperature dependent so that it will readily also be visible at room temperature for appropriately designed samples with both parallel and antiparallel alignment at the desired resonance field.

**Discussion**

To understand our findings of a sizable magnon spin valve effect in our system, we have to consider both alignment-dependent SC transport, as well as spin-to-charge conversion effects. Focusing first on the magnonic SC transport, we note that the induced resonant precession of the YIG film emits a magnonic SC traveling across the insulating AFM[34,35]. In very thin AFMs this is even possible for evanescent modes[36] (see Supplementary Note 3, where spin model simulations of FM/AFM bilayers and FM/AFM/FM trilayers are presented). The SC emitted by the YIG

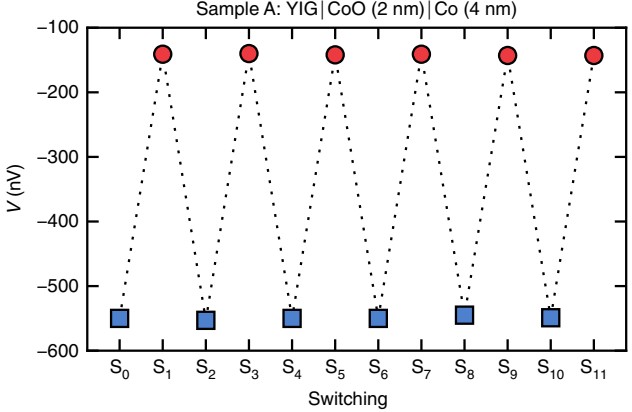

**Fig. 4** Alternating voltage levels by switching the Co magnetization. Total voltage for parallel (red circles) and antiparallel (blue squares) alignment of $Y_3Fe_5O_{12}$ (YIG) and Co measured in sample A at ferromagnetic resonance $H_{FMR}^{YIG} = 788$ Oe at $T = 120$ K. The presented sequence Si corresponds to the magnetization of the Co layer being switched repeatedly by driving the external field above (below) the Co coercive field $H_c^+ = 1250$ Oe ($H_c^- = -970$ Oe). The absolute voltage difference is $(408 \pm 5)$ nV and the relative change amounts to $(290 \pm 4)$ %

layer enters the CoO for both orientations of the YIG with equal probability (note that the CoO does not change its orientation in our experiments as confirmed by probing opposite field-cooling directions). The dispersion relation of the AFM has two branches. Magnons in each branch carry opposite spin angular momentum and have opposite polarization, which means that the magnetic moments in the AFM precess in opposite directions. This allows for an equal transmission of magnons through the AFM for both orientations of the YIG. In ferromagnets, however, magnons have a unique polarization such that only in the configuration where the two ferromagnets are parallel magnons from one can enter the other. For antiparallel orientation, they are reflected at the second interface (see Supplementary Note 3 for details). This effect alone can produce a purely magnonic spin valve effect. However, for the rather simple models simulated here, we find a larger SC transmissivity for parallel alignment of the ferromagnets. In contrast to this, we observe a stronger SC transport signal amplitude for the antiparallel orientation of the two ferromagnets. Though more realistic spin models might yield different results, we note that we cannot model the real system due

to the complexity of YIG and the unknown YIG|CoO interface properties. Consequently, we cannot check the potential impacts of the complex SC spectrum and interface transmissivity.

For a better understanding of our experimental observations, electronic effects, such as a spin-dependent ISHE in the Co layer or a spin-dependent effective spin transmissivity at the CoO|Co interface, should be discussed in addition to the magnonic effects in our sample. The magnonic SC flowing across the CoO-layer generates a spin accumulation in the Co-layer that results in a pure SC flow of spin-up and spin-down electrons moving in opposite directions. Due to the ISHE, electrons of both spin polarization are scattered toward the same direction, generating an electrical voltage measured in the experiments. Non-magnetic metals like Pt are spin-unpolarized and regarding their properties electrons of both polarizations are equal. A sign change of the SC realized by switching the YIG magnetization yields a sign change of the electrical voltage, but the amplitudes are similar, as shown in YIG|CoO|Pt multilayers by Qiu et al.[33]. Ferromagnetic metals like Co, however, are spin-polarized and exhibit significant differences in the intrinsic properties of spin-up and spin-down electrons. The measured ISHE voltage thus results from magnons coupling to two distinct electron channels, which can be described by a two-fluid model in the case of negligible spin-intermixing processes. Eventually, reversing the relative alignment of the YIG and Co magnetization manifests in a reversal of the SC polarization or, likewise an interchange of the roles of spin-up and spin-down electrons in the Co.

Electronic effects thus may explain the observation of asymmetric SC transmission voltages for the different alignments between YIG and Co with $V_{sc}^{\uparrow\downarrow} > V_{sc}^{\uparrow\uparrow}$. Since the intrinsic properties of spin-up and spin-down electrons are material specific, higher voltages for the parallel alignment of the ferromagnets could also be observed. Note that our observations for magnonic SCs are distinct from previous work, where when investigating thermal SC propagation in YIG|Cu|Co, which includes a non-magnetic metal as spin-conduit and thus electronic instead of magnonic spin injection in the Co layer, no significant change in the signal was reported[46]. Moreover, the concept of the presented experiment differs from previous reports on FMR spin pumping in YIG|Au|Py structures, in which a SC detection scheme comparable to that of non-local spin valves is implemented[47].

In conclusion, we demonstrated the magnon spin valve effect in YIG|CoO|Co multilayers by showing that the detected SC signal amplitudes depend on the relative alignment of YIG and Co. The total voltage signal measured includes two contributions, whose signs depend individually on the YIG and Co magnetization directions and thus yield different signal amplitudes and signs. This enables one to encode even two bits of information in the magnetic configuration of the spin valve. The presented setup gives an insight into spin-dependent effects in magnetic multilayers and provides a switch component at the SC detection site, thus making the work a key step toward further magnon-based logic gate operation.

## Methods

**Sample preparation**. Single crystalline YIG ($d \approx 5\,\mu$m) was grown by liquid-phase epitaxy and cut into samples of size 2 mm × 3 mm to ensure equal bulk properties. To promote the growth of the CoO layer, the YIG surface was optimized by means of a rapid thermal annealing process (see Supplementary Note 4). In an infrared furnace, YIG samples were pairwise arranged face-to-face and heated up to 1173 K at a heating rate of 50 K s$^{-1}$. The final temperature was kept for 30 min, before the samples were cooled down quickly to room temperature again. As a result, terrace like, smooth YIG surfaces of ≈1.11 Å roughness were obtained. Subsequent to the annealing procedure, CoO was deposited by reactive magnetron sputtering from a Co target in an Ar/O atmosphere employing a QAM-4-STSCP (ULVAC Inc.) system. During the deposition process, the substrate temperature was kept at 723 K. X-ray spectroscopy revealed a CoO growth along the [111] direction on top of annealed YIG samples (see Supplementary Note 4). Eventually, Co top layers were

grown in-situ at room temperature by non-reactive sputtering from the same Co target. Here, thickness combinations of $d_{CoO}/d_{Co} = 2$ nm/4 nm, 3 nm/4 nm, and 5 nm/6 nm were realized.

**Experimental setup**. Thermoelectric as well as FMR spin pumping measurements were performed in a physical property measurement (PPMS) Dynacool system, quantum design, Inc., which allows for temperature-dependent measurements from 10 to 400 K. High temperatures were avoided to prevent undesired degradation of the sample stack. For the ANE measurements the samples were clamped in between two aluminum nitride plates with high thermal conductivity. The top plate is heated by means of a resistive chip heater, whereas the bottom plate serves as a heat sink. The resultant out-of-plane temperature gradient induces, in the presence of an external magnetic field, the ANE thermovoltage, which is detected by a nanovoltmeter.

For the spin pumping experiments the samples were attached to a copper-based coplanar waveguide (CPW) with the Co layer facing the CPW. The sample fixation, as well as the electrical insulation of the Co from the copper is given by a 10 μm thick double-sided tape. FMR of the YIG layer was achieved by feeding pulsed microwaves in the CPW while an external magnetic field is applied. The frequency of applied microwaves is $f = 4.5$ GHz with a typical applied microwave power of $P = 23$ dBm. The respective spin-pumping signal was detected using a lock-in amplifier, which was triggered by the microwave source.

**Data availability**. All relevant data are available from the corresponding author on request.

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

## Acknowledgements

This work was supported by the Deutsche Forschungsgemeinschaft (DFG) (SPP 1538 Spin Caloric Transport, SFB767 in Konstanz, and SFB TRR173 SPIN+X in Mainz), the Graduate School of Excellence Materials Science in Mainz (DFG/GSC 266), the EU project INSPIN (FP7-ICT-2013-X 612759), the ERATO Spin Quantum Rectification Project (No. JPMJER1402) from JST, Japan, Grant-in-Aid for Scientific Research on Innovative Area Nano Spin Conversion Science (No. JP26103005), the Grant-in-Aid for young scientists (B) (No. JP17K14331) from JSPS KAKENHI, Japan, and the Center of Excellence QuSpin (project number 262633) funded by the Research Council of Norway. We thank the DAAD SpinNet and MaHoJeRo projects for supporting the Tohoku-Mainz collaboration and M.K. thanks ICC-IMR at Tohoku University for their hospitality during a visiting researcher stay at the Institute for Materials Research. T.K. is supported by JSPS through a research fellowship for young scientists (No. JP15J08026). Z.Q. acknowledges support from the Fundamental Research Funds for the Central Universities (DUT17RC(3)073).

## Author contributions

M.K. and U.N. proposed the study, which was refined and supervised additionally by E.S. The samples were fabricated by Z.Q., T.N., J.C. and F.F. with the help of M.K. The measurements were carried out by J.C., F.F. and M.K. with the help of D.H. and R.R. The data were analyzed by F.F. and J.C. The theoretical work was performed by U.R. and V.G. with supervision from U.N. and J.S. All authors participated in the discussion and interpreted results. J.C. drafted the manuscript with the help of M.K. and all authors commented on the manuscript.

## Additional information

**Competing interests:** The authors declare no competing interests.

