## [Peer Review File · Nature Communications]

PEER REVIEW FILE

Reviewers' comments:

Reviewer #1 (Remarks to the Author):

The paper is devoted to investigation of YIG/CoO/Co system under influence of external microwave pumping. By applying microwave radiation, the authors excite FMR in YIG and observe a dc-voltage on Co-film, which shows a contribution, depending on the mutual orientations of the magnetizations in YIG and Co. Although the finding is interesting, I believe that the paper possesses two important shortcomings, which hinder its publication in the present form.

1. The finding is put in the wrong context.

a) One reads in the introduction, that "...the entity of a great number of magnons with equal propagation direction yields a pure spin current carrying information in the form of angular momentum." Later one finds "an extremely advantageous functionality would be the integration of a switch and the detection in a single device analogously to the GMR-based spin valve effect". Thus, magnon spin-valve effect should be connected with propagating spin waves. In the experiment, however, non-propagating, FMR mode has been investigated.

b) If one considers a classical electronic spin-valve effect, it is connected with a resistance to electric current, which depends on the mutual orientation of the magnetizations of the layers building the valve. Thus, to demonstrate a magnon spin-valve effect, one should show different values of "magnon" resistance. It has not been done in the described experiments. Instead, even if one follows the author's interpretation, a detector for FMR with a variable sensitivity is realized.

c) Contrary to electrons, spin waves strongly interact with magnetic fields created by magnetic layers. Therefore, the flow of spin waves and the corresponding pure spin current can be easily controlled by external magnetic fields (K. Vogt et al., Nature Comm., 5 (2014) 3727) or even by remanent magnetization (K. Wagner et al., Nature Nanotech., 11 (2016) 432.) as a reconfigurable device. Thus, the task formulated by the authors has been already realized.

2. There is an essential drawback in the experimental scheme and/or interpretation of the results. In fact, if one measures the electric resistance of a device by recording the voltage, one should be sure that the corresponding current is constant. For the current experiment, it means that the dc-

voltage on Co should be measured while the amplitude of the FMR-precession in YIG is constant. I have found no indication that this amplitude was kept constant when the authors varied the orientations of the magnetization. #however, this effect cannot be excluded a priori, in particular, if one keeps in mind the authors themselves discussed thermally induced effects due to dynamic microwave heating. It is well known that due to relatively low critical temperature of YIG its microwave properties are strongly temperature dependent

Reviewer #2 (Remarks to the Author):

Research efforts addressing magnonic spin currents (or spin waves) for data transmission and processing have increased tremendously in recent years. In this manuscript, the authors took use of a delicate YIG/CoO/Co trilayer structure with YIG acting as the spin current source excited through FMR spin pumping, Co acting as the spin current detector, whereas the antiferromagnetic CoO spacer allowing the propagation of the spin current, and meanwhile inducing large coercivity for the Co layer to even exceed the FMR resonance field of the YIG, thus they found that the detected spin current transmission signal amplitudes strongly depend on the relative alignment of the YIG and Co magnetization directions. The appreciable magnon spin valve effect is very interesting and timely for the magnonics community, therefore the manuscript could be recommended to publish in Nature Communications. However, two points about the result have to be clarified before publication.

(1) The authors only considered the exchange biasing effect between CoO and Co layers, completely ignored the similar effect between YIG and CoO. Both should occur simultaneously. The result discussion without considering this point is very problematic. Since the YIG film is thick in micrometers, far beyond its exchange length, one could not observe M-H loop shift and/or coercivity enhancement for the YIG layer in the present samples, but it does not mean that there is no exchange interaction between YIG and CoO. In fact, it is highly possible that the exchange spring is formed around YIG interface for the antiparallel magnetization configuration, which will greatly alter the spin current behavior.

(2) Minor point, Page 10, Magnon spin valve effect: Since the switching field of the Co layer is higher than the FMR resonance field, and the result in figure 4 was obtained by switching the Co magnetization rather than switching the YIG magnetization, it is better to give the switching field value in this measurement.

Reviewer #3 (Remarks to the Author):

In the present manuscript, the authors have reported the spin valve effect in Yttrium Iron Garnet

(YIG)/CoO/Co multilayers. By irradiating microwave to YIG, a ferromagnetic resonance (FMR) is induced. As a result of the dumping effect, a spin current is generated and transported to the Co layer through an antiferromagnetic CoO layer. It is then converted to a charge current via the inverse spin Hall effect in Co and detected as a dc voltage. The detected dc voltage depends on the magnetic configurations of YIG and Co: it is about 0.5 uV at the FMR resonance frequency for the antiparallel state, while it becomes small (0.1 uV) for the parallel state. The result is similar to the spin valve effect and the obtained voltage ratio is more than 100%, but compared to conventional devices, in the present case, magnon spin current mediates spin information. They have also realized the switching of the parallel and antiparallel states with a fixed magnetic field.

In the following I raise several questions.

- (1). The magnetic switching fields of Co are determined from the anomalous Nernst effect in Fig. 1b, but they are quite different from those in Ref. 25. What is the reason for this difference? The difference of H_{c^+} (1250 Oe) and H_{c^-} (-970 Oe) is explained by the exchange bias of CoO, but I wonder if these large switching fields can also be explained by the exchange bias.
- (2). In Figs. 1c-1f, the detected voltage is decomposed into two symmetric parts. One comes from the ISHE in the Co layer and the other is due to heating or spin rectification effects. I cannot understand well why the latter generates the symmetric voltage signal. It would be nice if the authors could add some intuitive explanations on page 7 (instead of simply citing Ref. 36). Even in the case of YIG/CoO/Pt in Ref. 30, the observed voltage signal looks not so perfectly symmetric. If Co is replaced by Pt in the present case, does the V_{sr} signal disappear? At least there should be no sign change and much less temperature dependence. Have the authors checked V_{sr} in YIG/CoO/Pt?
- (3). Why does the spin rectification voltage V_{sr} suddenly disappear at around 150 K?
- (4). If CoO is replaced by other simple insulators such as MgO, does the dc voltage completely disappear?
- (5). As for the switching experiments, I could not understand how the magnetization switching has been realized. What is the important parameter for this switching experiment (what is the difference in each event S_0, S_1, \dots)? More detailed explanations should be added in the manuscript or supplementary information.
- (6). The magnon contribution should be quite different below and above the Neel temperature T_{Neel} . Especially, above T_{Neel} , there is no magnetic ordering and thus CoO becomes a simple insulator. In this situation, the coherence length for pure spin current should be very short.

Nevertheless, the detected dc voltage does not show a large reduction above T_{Neel} . What is the reason for this temperature dependence? What is the typical coherence length for pure spin current above T_{Neel} ?

Responses to Reviewers' Comments:

Reviewer #1 (Remarks to the Author):

The paper is devoted to investigation of YIG/CoO/Co system under influence of external microwave pumping. By applying microwave radiation, the authors excite FMR in YIG and observe a dc-voltage on Co-film, which shows a contribution, depending on the mutual orientations of the magnetizations in YIG and Co. Although the finding is interesting, I believe that the paper possesses two important shortcomings, which hinder its publication in the present form.

We thank the reviewer for her/his consideration of the manuscript and that she/he appreciates that the presented work is interesting. In the following, we will respond to the reviewer's concerns to overcome the critical points mentioned and make the paper apt for publication.

1. The finding is put in the wrong context.

a) One reads in the introduction, that "...the entity of a great number of magnons with equal propagation direction yields a pure spin current carrying information in the form of angular momentum." Later one finds "an extremely advantageous functionality would be the integration of a switch and the detection in a single device analogously to the GMR-based spin valve effect". Thus, magnon spin-valve effect should be connected with propagating spin waves. In the experiment, however, non-propagating, FMR mode has been investigated.

b) If one considers a classical electronic spin-valve effect, it is connected with a resistance to electric current, which depends on the mutual orientation of the magnetizations of the layers building the valve. Thus, to demonstrate a magnon spin-valve effect, one should show different values of "magnon" resistance. It has not been done in the described experiments. Instead, even if one follows the author's interpretation, a detector for FMR with a variable sensitivity is realized.

c) Contrary to electrons, spin waves strongly interact with magnetic fields created by magnetic layers. Therefore, the flow of spin waves and the corresponding pure spin current can be easily controlled by external magnetic fields (K. Vogt et al., Nature Comm., 5 (2014) 3727) or even by remanent magnetization (K. Wagner et al., Nature Nanotech., 11 (2016) 432.) as a reconfigurable device. Thus, the task formulated by the authors has been already realized.

We thank the referee for mentioning these apparently unclear points and the revisions of the manuscript have allowed us to clarify these as detailed below. Furthermore we have carried out significant further experiments in response to the questions that have allowed us to unambiguously show that the observed effect is not limited to the geometry probed in the original work in the manuscript.

First of all we agree that the aforementioned works by Vogt *et al.* and Wagner *et al.*, together with the magnon transistor introduced by Chumak *et al.* (Nat. Commun. 5, 4700, 2014), mark key steps towards the realization of magnon spintronics/magnon logic. The general difference between these approaches and our work, as remarked correctly by the reviewer, is that in our case the logic step is implemented at the detection side of the spin currents. We would like to point out, however, that the term spin valve is very broadly used to describe effects of spin transport that depend on the

relative orientation of two ferroic layers. This includes obviously GMR and TMR, but also includes antiferromagnetic and ferroelectric layers [Quindreau *et al.*, *Sci. Rep.* **5**, 9749 (2015)] and even superconductors. However, the term spin valve is not strictly reserved to these mechanisms and even single magnetic layer effects have been called spin-valves or spin-valve like [Yanson *et al.*, *Low Temp. Phys.* **39**, 279 (2013); Gould *et al.*, *Phys. Rev. Lett.* **93**, 117203 (2004); Park *et al.*, *Nat. Mater.* **10**, 347-351 (2011)]. Finally, single molecule effects have also been termed as spin valves [Urdampilleta *et al.*, *Nat. Mater.* **10**, 502-506 (2011)], where very different underlying physical principles act. Such systems could be called also detectors with variable output signal as the referee has rightly pointed out. So terminology is more of a semantic question and while GMR and TMR are (relatively) well-defined terms with a narrow use based on the physical mechanism, the term “spin valve” is used in the literature in a broad sense as a more generic term. It can be used for different underlying physical mechanisms that lead to different outputs depending on the alignment of magnetic elements. In the revised manuscript we have tried to make this clear and tone down the analogies to the GMR effect. If the referee feels that we should further adjust the terminology, description and explanation to make this even clearer, we are very open to concrete suggestions.

So we report a (to our knowledge) novel physical effect that constitutes a spin valve effect based on an insulating system where the spin current is carried by magnons.

Concerning spin current propagation, FMR spin-pumping indeed induces spin currents as, for instance, visible in the simply YIG/Pt geometry. There only a signal is visible, if a spin current is flowing in the Pt in the direction away from the YIG/Pt interface. So there is a propagating spin current in the Pt that results from the FMR in the YIG. Analogously, we excite the FMR mode in the YIG layer, and in accordance with the models presented by Rezende *et al.* [*Phys. Rev. B* **93**, 054412 (2016)] and Chen *et al.* [*Phys. Rev. B* **94**, 054413, (2016)] the YIG magnetization precession emits a magnonic spin current into the insulating antiferromagnet, here CoO, that is carried by diffusive thermal magnons (at temperatures $>0\text{K}$) or possibly evanescent modes [Khymin *et al.*, *Phys. Rev. B* **93**, 224421 (2016)]. This mechanism is furthermore not limited to FMR excitations, but is also valid for the transmission of thermal magnon currents generated directly in the ferromagnet by the spin Seebeck effect [e.g. for stacks like YIG/NiO/Pt, see Lin *et al.* *Phys. Rev. Lett.* **116**, 186601 (2016); Prakash *et al.* *Phys. Rev. B* **94**, 014427 (2016)]. In our case this thermal spin current was not detected, as the SSE signal is up to a factor 100 smaller than the spin pumping signal (for the typically achievable temperature gradients and typical spin pumping powers). Finally, in the metallic ferromagnet the spin current continues as a pure spin current by electron diffusion (analogously to the spin current in the Pt in the YIG/Pt case). Furthermore, recently an intermixing between electron and magnon spin currents in itinerant ferromagnets have even been predicted due to strong exchange interaction so that likely the spin current is carried in the full stack at least partially as propagating magnons [Cheng *et al.* *Phys. Rev. B* **96**, 024449 (2017)].

[REDACTED]

[REDACTED]

[REDACTED]

In conclusion, we find that (a) although the FMR mode is used in our work in the manuscript, propagating spin waves are injected in the AFM and the Co layer, (b) the observed effect is universal and not exclusive to FMR detection, and (c) to obtain a spin valve effect in magnetic insulators different physical mechanisms can be used. We present a novel one in a system where the spin current is carried by magnons here, which we feel warrants the term “spin valve”.

Nonetheless, we are of course always open to concrete suggestions of how to revise the wording of the text to make the manuscript clearer.

2. There is an essential drawback in the experimental scheme and/or interpretation of the results. In fact, if one measures the electric resistance of a device by recording the voltage, one should be sure that the corresponding current is constant. For the current experiment, it means that the dc-voltage on Co should be measured while the amplitude of the FMR-precession in YIG is constant. I have found no indication that this amplitude was kept constant when the authors varied the orientations of the magnetization. #however, this effect cannot be excluded a priori, in particular, if one kepps in mind the authors themselves discussed thermally induced effects due to dynamic microwave heating. It is well known that due to relatively low critical temperature of YIG its microwave properties are strongly temperature depended

We thank the reviewer for bringing up this point allowing us to provide an additional check to further corroborate our results and interpretation. During all experiments performed, except for the power-dependent measurements of course, the applied microwave power has been kept constant at $P_{\text{applied}} = 23$ dBm. In our understanding and to our best knowledge, a significant change of the FMR-precession amplitude results in a measurable difference in the microwave power absorbed by the ferromagnet. Below we compare in Fig. 2a-d (also in the revised supplementary information as Fig. S6) the microwave absorption spectra in sample A [YIG (5 μm)/CoO (2 nm)/Co (4 nm)] for both parallel and antiparallel alignment as well as for positive and negative external magnetic field. The data reveal that for all cases the absorbed microwave power at resonance is very close to 50 mW. Within the error bars, the absorbed power is thus identical and small differences (<1%) are likely to result from uncertainties in the field calibration and read-out leading for instance to a typical bi-polar difference signals as shown for example in Fig. 2 (c). The minor deviations observed (Fig. 2c,d) are in any case small compared to the 120 % change of the spin pumping amplitude V_{sc} (see page 11 in the main manuscript) and could be caused by inaccuracies of the external field read-out. We are aware of the fact that the absorption spectra involve the whole bulk system so that changes of the rotation amplitude at the YIG/CoO may be below the available resolution. However this is not the case, as we see from the observed sensitivity of the FMR linewidth towards minor changes of the CoO thickness and thus the spin pumping effect at the YIG/CoO interface (see Fig. S5 in the supplemental information).

As this point is indeed relevant, the respective data have been added to the supplemental information of the manuscript (section S3.2, new Fig. S6).

For the temperature dependent change of the YIG microwave properties, considering the equal power absorption we would naturally assume symmetric heating effects. Thus, if the properties should change, they should change for parallel as well as antiparallel alignment in the same manner and therefore not yield an impact on the qualitative nature of our observations.

Figure 2 (a),(b) Absorption spectra of sample A [YIG(5 μm)/CoO(2 nm)/Co(4 nm)] obtained for microwave irradiation ($f = 4.5$ GHz, $P_{\text{applied}} = 23$ dBm) as a function of magnetic field for parallel and antiparallel alignment of YIG and Co magnetization, measured at $T = 120$ K. (c),(d) Difference of microwave absorption for parallel and antiparallel alignment of YIG and Co from (a),(b). Relative differences of up to 1 % are observed, which can result from small field shifts that originate from the uncertainty due to finite field step resolution used in the experiment.

Reviewer #2 (Remarks to the Author):

Research efforts addressing magnonic spin currents (or spin waves) for data transmission and processing have increased tremendously in recent years. In this manuscript, the authors took use of a delicate YIG/CoO/Co trilayer structure with YIG acting as the spin current source excited through FMR spin pumping, Co acting as the spin current detector, whereas the antiferromagnetic CoO spacer allowing the propagation of the spin current, and meanwhile inducing large coercivity for the Co layer to even exceed the FMR resonance field of the YIG, thus they found that the detected spin current transmission signal amplitudes strongly depend on the relative alignment of the YIG and Co magnetization directions. The appreciable magnon spin valve effect is very interesting and timely for the magnonics community, therefore the manuscript could be recommended to publish in Nature Communications. However, two points about the result have to be clarified before publication.

We thank the reviewer for her/his appreciation that our work is fundamentally appropriate for publication in Nature Communications as well as her/his time and effort spent to evaluate it. We furthermore thank for sharing her/his concerns, to which we will respond in the following so that the revised manuscript becomes apt for publication.

(1) The authors only considered the exchange biasing effect between CoO and Co layers, completely ignored the similar effect between YIG and CoO. Both should occur simultaneously. The result discussion without considering this point is very problematic. Since the YIG film is thick in micrometers, far beyond its exchange length, one could not observe M-H loop shift and/or coercivity enhancement for the YIG layer in the present samples, but it does not mean that there is no exchange interaction between YIG and CoO. In fact, it is highly possible that the exchange spring is formed around YIG interface for the antiparallel magnetization configuration, which will greatly alter the spin current behavior.

We agree that an exchange bias exerted by the CoO layer at both the YIG/CoO and the CoO/Co interface is a potential influence on the detected spin current signal. To check for this possibility, we repeated measurements with the sample being field-cooled in external fields of switched polarity ($H_{\text{ext}} = \pm 90$ kOe) to change the exchange bias direction. While the exchange bias for the Co layer reverses when one reverses the field cooling direction, the resultant data show no significant difference for both the detected spin signal and the microwave absorption spectra. A major impact of the exchange bias effect on the spin current transmissivity of the CoO/Co interface should alter the spin signal in a measurable manner, which is not observed. This finding is furthermore corroborated by the symmetry of the voltage signals for positive/negative fields as shown in Fig. 1 of the main manuscript. With regard to the microwave absorption spectra, considering the sensitivity of the absorption peak linewidth towards the thickness of the CoO/Co bilayer shown in Fig. S5 in the supplemental information, a difference in the linewidth would be expected if a potentially formed exchange spring would influence the spin pumping efficiency at the YIG/CoO interface. Finally, any exchange spring effect would yield a different behavior at a different applied field strength. We have changed the spin pumping frequency leading to different applied fields for the experiment and the results are all consistent showing that a change of the applied field has no significant influence on the signal and therefore there is no large exchange spring effect on the emitted spin current. Altogether, the data show that although it might be present at the CoO/YIG interface, the exchange bias effect does not influence to a significant degree the observed effect.

The respective information has been added to the manuscript (page 8, top).

(2) Minor point, Page 10, Magnon spin valve effect: Since the switching field of the Co layer is higher than the FMR resonance field, and the result in figure 4 was obtained by switching the Co magnetization rather than switching the YIG magnetization, it is better to give the switching field value in this measurement.

We thank the reviewer for this valuable comment. The respective information as well as a more detailed description of the switching and measurement process have been added to the main text of the manuscript and the caption of Fig. 4 (page 11).

Reviewer #3 (Remarks to the Author):

In the present manuscript, the authors have reported the spin valve effect in Yttrium Iron Garnet (YIG)/CoO/Co multilayers. By irradiating microwave to YIG, a ferromagnetic resonance (FMR) is induced. As a result of the dumping effect, a spin current is generated and transported to the Co layer through an antiferromagnetic CoO layer. It is then converted to a charge current via the inverse spin Hall effect in Co and detected as a dc voltage. The detected dc voltage depends on the magnetic configurations of YIG and Co: it is about 0.5 uV at the FMR resonance frequency for the antiparallel state, while it becomes small (0.1 uV) for the parallel state. The result is similar to the spin valve effect and the obtained voltage ratio is more than 100%, but compared to conventional devices, in the present case, magnon spin current mediates spin information. They have also realized the switching of the parallel and antiparallel states with a fixed magnetic field.

We thank the reviewer for her/his positive evaluation of our manuscript as well as sharing her/his concerns in a detailed manner. In the following, we will answer to the questions raised.

In the following I raise several questions.

(1). The magnetic switching fields of Co are determined from the anomalous Nernst effect in Fig. 1b, but they are quite different from those in Ref. 25. What is the reason for this difference? The difference of H_c^+ (1250 Oe) and H_c^- (-970 Oe) is explained by the exchange bias of CoO, but I wonder if these large switching fields can also be explained by the exchange bias.

We thank the referee for this important point, which has a simple explanation: The enhanced switching fields of the Co layer in the investigated samples result from the interaction with the antiferromagnetic CoO layer. While the exchange bias effect itself expresses itself as a unidirectional anisotropy (leading to an asymmetric shift of the hysteresis loop), it is also well known that exchange bias systems exhibit a strongly increased coercivity (see for instance review in J. Phys. D: Appl. Phys. 33 (2000) R247–R268). Different models are used to explain this, and one model for instance relies on the presence of distinct, independent grains in the polycrystalline CoO layer yielding a further uniaxial anisotropy, as discussed by Stiles *et al.* [Phys. Rev. B **59**, 3722 (1999)]. When the magnetic field, which is applied in the exchange bias direction, is reduced, these grains apply torques in opposite directions that result in some parts of the ferromagnet-magnetization switching by clockwise or counterclockwise reversal nucleation. Thus, a potential barrier is generated, which yields the enhanced coercive fields. When the external magnetic field is perpendicular to the exchange bias direction, however, the torques show in the same direction and the barrier is reduced, which results in lower coercive fields. Respective observations in CoO/NiFe systems have been reported for instance by Ambrose *et al.* [Phys. Rev. B **56**, 83 (1997)]. With respect to our findings, this explanation is plausible considering the significant decrease of the Co coercive fields with increasing temperature, which is connected to a loss of antiferromagnetic order in CoO and thus a reduction in the exchange bias (see Fig. S3/S4 in the supplemental information).

The abovementioned information has been added to the main manuscript (page 4) as well as, in a more detailed manner, to the supplemental information (section S2).

(2). In Figs. 1c-1f, the detected voltage is decomposed into two symmetric parts. One comes from the ISHE in the Co layer and the other is due to

heating or spin rectification effects. I cannot understand well why the latter generates the symmetric voltage signal. It would be nice if the authors could add some intuitive explanations on page 7 (instead of simply citing Ref. 36). Even in the case of YIG/CoO/Pt in Ref. 30, the observed voltage signal looks not so perfectly symmetric. If Co is replaced by Pt in the present case, does the V_{sr} signal disappear? At least there should be no sign change and much less temperature dependence. Have the authors checked V_{sr} in YIG/CoO/Pt?

We thank the referee for raising this point and apologize for the apparently incomplete description of symmetric and asymmetric contributions to V_{sr} , which we now have restated in a more detailed manner in the manuscript and explain here: In metallic ferromagnets, which in contrast to our experiment are directly driven by ferromagnetic resonance, spin rectification (SR) indeed includes both a symmetric and asymmetric contribution around the resonance field. This is due to the fact that the phase between the rf current, which drives the rf field inducing the magnetization precession and generates the SR signal by coupling into the ferromagnet, and the magnetization precession changes across H_{FMR} . For AHE spin rectification the time average voltage signal is given by

$$\langle V_{AHE} \rangle = \frac{I_0 R_{AHE}}{2} \theta_1 \cos \varphi.$$

I_0 is the coupled rf current, R_{AHE} the anomalous Hall resistance and θ_1 the cone angle of magnetization rotation. The crucial factor here is φ , which is the abovementioned phase and in general can be field-dependent. Further evaluation of this equation (following Ref. 36 in our original manuscript) then yields the following:

$$\langle V_{AHE} \rangle(H) = V_{asymm} \frac{\Delta H(H-H_{FMR})}{(H-H_{FMR})^2 + \Delta H^2} + V_{symm} \frac{\Delta H^2}{(H-H_{FMR})^2 + \Delta H^2}.$$

We fitted our data using the respective form for the Co signal, which revealed that the asymmetric contribution is small as compared to the symmetric contribution and therefore was omitted in the subsequent evaluation (see Fig. 1 below). This was also done in order to limit the number of fit parameters and to obtain a robust quantitative measure of the signal amplitudes. Including this additional fitting would actually however not change the values significantly (see also Fig. 1 below). A potential explanation for this observation, as stated in the original manuscript, is the Co being driven off-resonance and furthermore probably coupling to several YIG modes, which thus may yield an arbitrary phase φ with weak field dependence.

The comparison of the amplitude of the spin current transmission signal, which in any case is the actual focus of this work, with and without asymmetric fit reveals changes of up to 5 % (in absolute numbers ≈ 13 nV), which are within the fitting errors and the errors considered with respect to the device accuracy.

In response to the suggestion of the referee, we checked FMR spin pumping in a sample in which Co was replaced by Pt [YIG(5 nm)/CoO(2 nm)/Pt(5 nm)], see Fig. 2 below. Pt as a paramagnetic material does not exhibit a spontaneous magnetization and therefore no spin rectification signal (V_{sr}) should appear. As clearly visible in Fig. 2, the signal resembles a single Lorentzian line shape without any further contribution, which matches the aforementioned expectation and corroborates our explanation.

Finally, we would like to emphasize again that the actual scope of this paper is the alignment-dependent efficiency of the spin current detection in YIG/CoO/Co. The appearance of the Co spin rectification signal is an additional finding, whose discussion is of course of importance (and which was expanded in the main text of the manuscript on page 7) but not the main focus of this work.

Figure 1 Voltage response of sample A [YIG(5 μm)/CoO(2 nm)/Co(4 nm)] towards microwave irradiation ($f = 4.5$ GHz, $P_{\text{applied}} = 23$ dBm) as a function of magnetic field for (a) parallel and (b) antiparallel alignment of YIG and Co magnetization, measured at $T = 120$ K. Here, the data were fitted applying a function that allows a symmetric as well as an asymmetric contribution to the Co-dependent signal. As compared to the symmetric part, the amplitude of the asymmetric signal is negligible.

Figure 2 Voltage response of a YIG(5 μm)/CoO(2 nm)/Pt(5 nm) sample towards microwave irradiation ($f = 4.5$ GHz, $P_{\text{applied}} = 23$ dBm) as a function of magnetic field, measured at $T = 300$ K. The data shows clearly a single Lorentzian line shape without an additional spin rectification contribution.

(3). Why does the spin rectification voltage V_{sr} suddenly disappear at around 150 K?

We are not sure if we understand the referee correctly as the signal does not disappear at 150 K. We assume the referee refers to Fig. 2 in the main manuscript? There it is not the V_{sr} signal that disappears at $T = 150$ K (for sample A) but the spin current transmission signal for antiparallel alignment of the YIG and Co magnetization. This is due to the fact that with increasing temperature the unidirectional anisotropy that enhances the switching fields of the Co layer (see our answer above to point (1)) is reduced, so that antiparallel alignment of YIG and Co at the YIG resonance field is not achievable anymore.

We apologize if the labelling is misleading, we now added a further statement in the text to clarify this finding.

(4). If CoO is replaced by other simple insulators such as MgO, does the dc voltage completely disappear?

We agree that the replacement of CoO by other insulators without magnetic order and thus short spin diffusion, e.g. MgO, states a likely step in the further investigation of the observed effects. The

measured spin transmission signal, which clearly only depends on the YIG magnetization, will vanish as shown previously [Wang *et al.* Phys. Rev. B **91**, 220410(R) (2015)]. Therefore, this kind of measurement will only contribute to the further investigation of the origin of the Co-dependent signal. At this point, we would like to stress again that the main thrust of this manuscript is the alignment dependent spin current detection efficiency. Consequently, respective measurements are of interest for follow-up studies in which the Co-dependent signal is investigated systematically, but at the moment go beyond the scope of this work.

(5). As for the switching experiments, I could not understand how the magnetization switching has been realized. What is the important parameter for this switching experiment (what is the difference in each event S_0 , S_1 , ...)? More detailed explanations should be added in the manuscript or supplementary information.

We apologize if the description of the experiment procedure used in the switching experiment is insufficient. The switching events S_i correspond to the Co magnetization being switched by repeatedly driving the external field above (below) the Co coercive field of $H_{c,+} = 1250$ Oe ($H_{c,-} = -970$ Oe). Subsequent to every switching event, the voltage response of the sample to the microwave irradiation is probed at the resonance field of the YIG film.

We thank the referee for noting this and as this information indeed is crucial it has been added to the main text and the caption of Fig. 4 to enhance the comprehensibility of the text.

(6). The magnon contribution should be quite different below and above the Neel temperature T_{Neel} . Especially, above T_{Neel} , there is no magnetic ordering and thus CoO becomes a simple insulator. In this situation, the coherence length for pure spin current should be very short. Nevertheless, the detected dc voltage does not show a large reduction above T_{Neel} . What is the reason for this temperature dependence? What is the typical coherence length for pure spin current above T_{Neel} ?

It is true that in normal insulators spin diffusion is short [e.g. 0.69 nm (0.18 nm) in $\text{Gd}_3\text{Ga}_5\text{O}_{12}$ (SrTiO_3), see Wang *et al.* Phys. Rev. B **91**, 220410(R), 2015], which one may lead to the assumption that above the antiferromagnetic phase transition the signal should reduce dramatically. However, large spin signals above the Néel temperature are not an exclusive observation made in this work but also have been shown by, for instance, Lin *et al.* (Phys. Rev. Lett. **116**, 186601, 2016) in YIG/NiO/Pt and Qiu *et al.* (Nat. Commun. **7**, 12670, 2016) in YIG/CoO/Pt. The explanation provided is based on the presence of magnetic correlations. By neutron scattering, it has been demonstrated that in NiO, which shares the same crystallographic and magnetic structure with CoO, short-range antiferromagnetic correlations are observed at temperatures well above the phase transition (up to $1.5 T_{\text{Neel}}$, see Chatterji *et al.* Phys. Rev. B **79**, 172403, 2009). Following the conclusions made by Lin *et al.*, AFM magnons with wavelengths shorter than this correlation length survive and thus can still transport angular momentum. This effect can explain the weaker-than-expected reduction of the spin pumping signal above T_{Neel} .

Reviewers' Comments:

Reviewer #1 (Remarks to the Author):

The authors have properly addressed all my point of criticism as the experiment is concerned. Therefore, I can support publication of the paper.

However, since the authors themselves admitted that their device is a magnons detector, and it does not control magnon propagation, I would suggest the title:

Magnon detector based on ferroic collinear multilayer spin valve.

Correspondingly, a slight modification of the text is necessary.

Reviewer #2 (Remarks to the Author):

The authors have improved the manuscript by considering my comments. Although exchange spring should present in an exchange coupled FM/AFM, it seems that the exchange coupling at the YIG/CoO interface is little and its effect on the emitted spin current from YIG is negligible for the current samples as the author claimed. Therefore, I would like to recommend it for publication in NC.

Reviewer #3 (Remarks to the Author):

The manuscript has been revised, based on the comments in the reviewer's report. Now I recommend the present manuscript for publication in Nature Communications.

Responses to Reviewers' Comments:

Reviewer #1 (Remarks to the Author):

The authors have properly addressed all my point of criticism as the experiment is concerned. Therefore, I can support publication of the paper. However, since the authors themselves admitted that their device is a magnons detector, and it does not control magnon propagation, I would suggest the title:

Magnon detector based on ferroic collinear multilayer spin valve.

Correspondingly, a slight modification of the text is necessary.

We thank the reviewer for her/his appreciation of our work and for her/his support for publication of the manuscript in Nature Communications.

Following the suggestion made, we changed the manuscript title to *Magnon detection using a ferroic collinear multilayer spin valve*. Additionally, we now provide a statement on the definition of the term *magnon spin valve effect* used in the manuscript as being the alignment-dependent detection efficiency of magnonic spin currents in respective spin valve structures (page 3).

Reviewer #2 (Remarks to the Author):

The authors have improved the manuscript by considering my comments. Although exchange spring should present in an exchange coupled FM/AFM, it seems that the exchange coupling at the YIG/CoO interface is little and its effect on the emitted spin current from YIG is negligible for the current samples as the author claimed. Therefore, I would like to recommend it for publication in NC.

We thank the reviewer for her/his valuable time spent on the evaluation of our manuscript and for her/his recommendation for publication in Nature Communications.

Reviewer #3 (Remarks to the Author):

The manuscript has been revised, based on the comments in the reviewer's report. Now I recommend the present manuscript for publication in Nature Communications.

We thank the reviewer for her/his appreciation of the revised manuscript and for her/his recommendation for publication in Nature Communications.